# Expanding research impact through engaging the maker community and collaborating with digital content creators

Jacob L. Sheffield, Bethany Parkinson *, Aliya Bascom, Terri Bateman, Spencer Magleby, Larry L. Howell

Compliant Mechanisms Research Group, Department of Mechanical Engineering, Brigham Young University, Provo, UT, United States of America

* bethanyparkinson@gmail.com

**Data Availability Statement:** All relevant data are within the manuscript and its Supporting information files.

## Abstract

This paper proposes a method for increasing the impact of academic research by providing materials for public use, thus engaging the maker community, and by collaborating with internet content creators to extend the reach. We propose a framework for engagement and report a multi-year study that evaluates short, intermediate, and long-term outcomes, with a second effort to demonstrate repeatability of the short-term outcomes. In the first study, we posted forty-one 3D printable compliant mechanisms on public repositories and collaborated with physicist and content creator Derek Muller (Veritasium YouTube channel). Outputs and outcomes from this interaction were measured over 3 years. The framework was exercised again with four new 3D printable mechanisms in collaboration with engineer and STEM influencer Mark Rober. The proposed methods aim to help researchers extend the reach of their work to broader audiences, including professional engineers, hardware designers, educators, students, researchers, and hobbyists. This work demonstrates promising impacts of the framework, including (1) extending public awareness of research findings to broader audiences by engaging the maker community and collaborating with content creators, (2) accelerating the pace of innovation and further hardware-based research through public application of research findings, (3) fostering a culture of open-source design and collaboration among other researchers, engineers, educators, and makers, and (4) increasing utilization of peer-reviewed published content. These outreach practices can be valuable tools for researchers to increase impact of and excitement for their research.

## Introduction

Engineering research is often directed to research peers. However, researchers and their sponsors often desire for their results to make an impact beyond their peers. To the general public, research papers can be inaccessible and difficult to understand. Disseminating research generally requires the research results to be translated into formats that are accessible to broader audiences. In this paper, we propose that engaging the maker community through online

**Funding:** This work was made possible by the National Science Foundation (NSF, https://www.nsf.gov/) through Award No. 1663345, "Mechanisms on Developable Surfaces", awarded to LLH. This award included as part of its Broader Impacts component a task to engage the maker community. The sponsors did not play any role in the study design, data collection and analysis, decision to publish, or preparation of the manuscript.

**Competing interests:** The authors have declared that no competing interests exist.

design-sharing platforms and collaborating with digital content creators such as STEM influencers can widen and accelerate the impact of engineering research. Outreach and education that employ multimedia to present engineering research findings and connect with the public in innovative ways, such as displaying dynamic and hands-on engineering research artifacts in museums [1] or offering immersive virtual reality (VR) experiences [2] facilitate knowledge transfer and expand research opportunities [3, 4]. The evolving media landscape [5] offers opportunities to enhance dissemination of research beyond what can be done solely through traditional academic publications. Technology, digital media, and the maker movement are transforming education [6] and offer excellent opportunities for researchers to disseminate their work in new ways. In this paper, we propose a framework for disseminating research to larger audiences and evaluate its effectiveness by exercising it in a case study. We measure the short, intermediate, and long-term outcomes of this case study, then repeat it and show similar short-term outcomes in a second case study. The history and motivation of the study were described previously by Howell and Bateman [7].

Sharing research results through contemporary avenues in parallel to traditional dissemination approaches can seed societal impact. Publicly providing open-source hardware (OSH) designs that can be 3D printed contributes hands-on tools for demonstrating key mechanical principles [8] and is a straightforward means for individuals to appreciate key concepts [9]. Engaging the maker community has untapped potential to advance scientific research and improve collaboration potential among researchers and the public [6, 10].

Furthermore, educational science videos on digital platforms are shown to facilitate learning by providing simplified explanations, visual demonstrations, and personalized content [11]. Individuals use platforms like YouTube to educate themselves about science-related topics [12, 13]. Many educators are adapting curriculum and learning materials to fit an evolving digital media landscape [14]. Researchers and the engineering academic community must also adapt. An equal effort in the preparation and dissemination of research is imperative to making broader impacts and finding future success.

Like other STEM outreach efforts, this work aims to engage, inspire, and educate the upcoming generation of scientists, engineers, and inventors [15]. The endmost goal of this outreach effort is to help broader audiences understand new technologies, employ greater creativity, and recognize how a specific technology can be applied in and expand various industries.

The content of this outreach project centers around a mechanical engineering research group that focuses on compliant mechanisms [16] and origami-inspired engineering [17]: the Compliant Mechanisms Research Group at Brigham Young University. This research is uniquely suited to demonstrate the functionality of mechanisms using 3D printable models. However, the concepts in this paper can be applied to various research types with varying degrees of modification.

## The maker movement

The maker community is emerging as an influential contributor to instructional content and product design to help accelerate the adoption of new technologies and fabrication practices [18]. Professional designers, engineers, hobbyists, educators, and students increasingly use OSH resources from design-sharing repositories [19–24] and rely on resources from content creators to supplement formal instruction such as textbooks and peer-reviewed publications. The maker movement is built on the collection of emerging technologies and practices [25], which enables users to collaborate and create new products [26, 27]. The movement represents a revolution in the product development process [28], with some referring to it as the third industrial revolution [29, 30].

The participation of both 'amateurs' and 'professionals' in the design process to produce OSH was exemplified during the COVID-19 global pandemic. Hundreds of designers participated in the rush to create critical open-source personal protective equipment (PPE) [31, 32]. The nature of open-source design-sharing platforms allowed for improved collaboration, continuous development, and design dissemination of PPE [33–36]. This trend in open-source hardware mirrors that of the open-source software (OSS) movement, which proved that a network of volunteers can write software code just as well as professional developers [37]. The OSH movement is similarly breaking down barriers between 'amateurs' and 'professionals'— and thus enabling a growing group of citizen scientists—as design software and rapid prototyping technology become more accessible [38].

Lindtner et al. contend that "we have to take seriously these maker practices, not just as a hobbyist or leisure practice, but as a professionalizing field functioning in parallel to research and industry labs" [26]. Members of the maker community apply their experience to work on pursuits ranging from recreational DIY (do-it-yourself) projects to sophisticated engineering undertakings and product design [39].

Makerspaces are public, shared-access collaborative workspaces found inside schools, public libraries, or private facilities to make, learn, explore, and share [40, 41]. They provide hands-on learning experiences through access to high-end manufacturing equipment and physical or digital prototyping resources. The scope and objectives of makerspace activities complement those of traditional education channels, such as schools [42], focusing on maker education and the notion of 'making' as a framework for learning [43]. The rapid growth of Makerspaces [44] and maker resources has been fueled by the availability and affordability of 3D printers, CNC machines, laser cutters, and other prototyping tools [45, 46]. Online design-sharing repositories, like Thingiverse or Printables, provide online easy-to-access models that help support makers and makerspaces.

Design-sharing repositories vary in content type and the community demographics they attract. Platforms may provide 3D printable models and instructions for DIY projects for makers or technical models for engineers, designers, and animators. Some platforms engage the maker community and foster open collaboration by hosting industry-sponsored design competitions [47]. With over 2.5 million 3D models, Thingiverse (Ultimaker) [48] has the most extensive online 3D model repository and 3D printing community at the time of writing [49]. Printables (Prusa Research) is also a popular repository, with over half a million models. Users share open-source designs under a Creative Commons license which grants copyright permissions for creative and academic work [50, 51].

Thingiverse was selected as the best option at the time of the first study to host the 3D printable models produced for this outreach experiment. Printables was added for the second study. Instructables, another popular online DIY design-sharing platform, hosts a wider variety of projects ranging from standalone 3D printable designs to complex instructional DIY projects. Instructables was used to host non-3D printable models, such as paper origami tutorials.

## Digital content creators

Educational content creators produce digital media to facilitate the transfer of knowledge [13]. While an academic research group may be familiar with effective distribution channels to disseminate technical research results to the academic community, they traditionally have had limited channels or resources to reach broader audiences. To disseminate research results and knowledge more effectively, a university research group can collaborate with educational content creators with a substantial audience on social platforms. While in this case the CMR lab

does not have a large following of its own, it is possible to create your own following and thus not have to rely on popular content creators to disseminate information.

## Proposed framework

Fig 1 outlines a suggested structure for sharing research results through modern distribution platforms. This process complements traditional publication goals and includes: (1) conducting technical research, (2) publishing a conference or journal article, (3) publishing summarized key-findings to an online blog associated with the lab website, (4) uploading modified engineering models and resources, such as 3D printable models, to online design-sharing repositories, (5) distributing quality media on social media platforms, (6) collaborating with appropriate digital content creators, and (7) continually engaging with the community through online interactions when possible. Following this framework can lead to more effective dissemination, increasing awareness among broader audiences and creating lasting long-term societal and economic impacts.

Monitoring engagement success and tailoring future outreach efforts to align with preferred channels and materials that resonate with broader audiences can optimize this process.

## Methods

Project activities that contributed to producing direct outputs are outlined in this section. See Fig A in S1 File for a summarized Project Logic Model. The impact of the project outputs was gauged by evaluating short, intermediate, and long-term outcomes. Short-term outcomes included traffic and engagement data from internal analytic tools on Thingiverse, Printables, and Instructables, CMR website traffic via Google Analytics, YouTube analytics of videos that are embedded on each of the maker repositories, and survey responses from Thingiverse visitors. Brigham Young University's IRB has approved the research study and survey as exempt level, category 4. The survey did not request any identifying information, and the data were analyzed anonymously. The need for a consent form was waived by the IRB. Consent was implied through the non-obligatory nature of the survey.

Intermediate outcomes were evaluated with organic searching on social media and internet appearances with key search terms, Google Alerts, textbook distribution, and other internally evaluated metrics. Long-term outcomes were inferred with a secondary survey conducted two and a half years later to the same individuals who completed the initial survey.

## Creating content

To create content that demonstrates this lab's research, we modified engineering 3D models from previous and ongoing research to allow for ease of 3D printing and demonstration. We then uploaded the models to design-sharing platforms periodically both before and after collaborating with each influencer. The models were selected to exemplify novel engineering principles, teach specific skills, or promote innovative thinking to solve engineering problems.

Candidates for upload were evaluated against the following criteria:

1. Does not infringe on prior intellectual property or contractual agreements with third-party sponsors

2. Can be simplified for 3D printing or ease of replication

3. Inspires creative problem-solving

4. Demonstrates or teaches a specific engineering or design principle

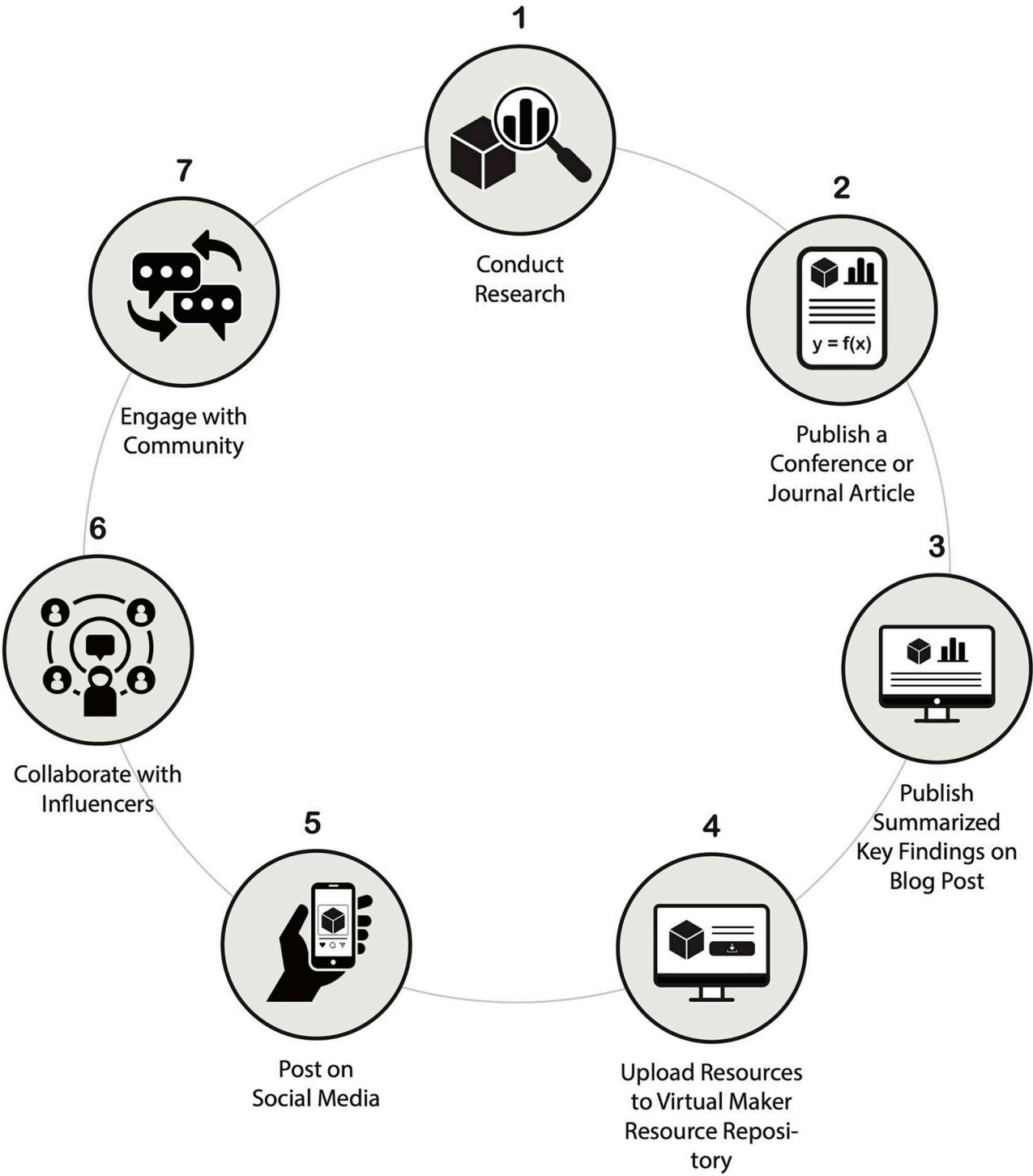

**Fig 1. Proposed framework for dissemination of research findings using contemporary distribution channels.**

Other considerations which may increase the likelihood of broader reach and positive reception include:

1. Follows current trends or themes (e.g., holiday applications or connections to current events)

2. Is tailored towards a specific content creator who can help promote the work

3. Has a practical application

For future studies, it would be beneficial to measure which of these considerations made a model more attractive to the maker community.

We optimized the designs to allow makers to intuitively replicate the models with only basic equipment, materials, and fabrication processes. This required factoring in material limitations of common user-friendly filaments, particularly PLA (polylactic acid), and resolution constraints of entry-level FDM 3D printers. Although in an actual implementation, a model may require advanced materials, manufacturing methods, and hardware to achieve performance consistent with the design intent, the simplified outreach models still have comparable overall function and provide a hands-on experience.

An example of modifying a complex design for 3D printing is a model for a constant-force mechanism shown in Fig 2. The original prototype was created to validate the initial research required various materials, hardware, and manufacturing methods. The modified design is one unique part that can be 3D printed with PLA on an entry-level 3D printer without the need for printed supports. The part is duplicated four times and assembled with compliant snap joints. Not including the print time, the active time it takes someone from downloading the part to assembling it is less than five minutes.

Models uploaded to design-sharing repositories were strengthened through the addition of quality media content such as high quality photos, animations, and videos. Generally, posts also included instructions for home printing, a description of the device, and learning opportunities. Linked references were also made to research articles showcasing applications, corresponding intellectual property, and supplementary reading material to encourage additional learning.

## Revised lab website content

The technical content that was uploaded to the online design-sharing repositories was simplified to a form that is more accessible to the general public. Resources were also provided

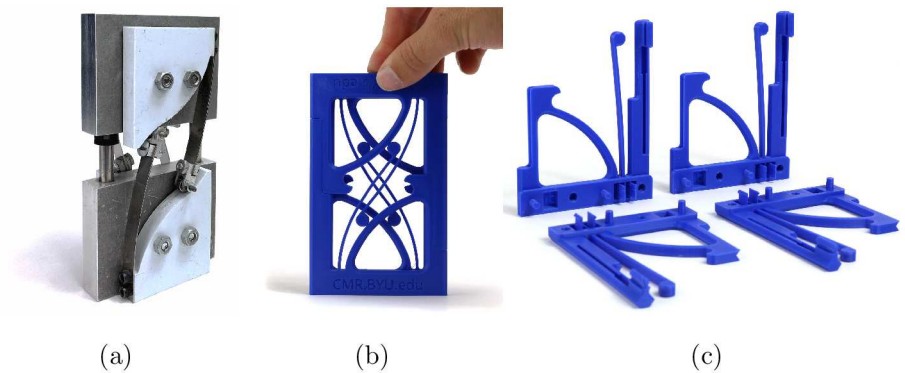

(a)                         (b)                         (c)

**Fig 2.** Comparison of (a) a constant-force mechanism model created for technical validation and inclusion in research publications, (b) a modified 3D printable and easy-to-assemble design for the maker community with (c) only one unique part replicated four times. The modified constant-force mechanism design requires less than five minutes of active attention from download to assembly.

through the posts on maker repositories and through the research lab's website (https://compliantmechanisms.byu.edu/maker-resources) to expand visitors' knowledge. These included technical publications, videos, and examples.

## Tutorials to facilitate sustainability

Since turnover of students in a typical university research lab is fairly high, tutorials and templates describing the process for prototyping and posting material onto maker spaces were created for the purpose of outlining a repeatable and sustainable process to help the next generation of students continue supporting the ongoing outreach strategy. They have been used extensively and to great benefit. The creation of such templates is recommended to create a cost-efficient and sustainable effort and to enable posted research to remain visually professional and cohesive to a general theme.

## Collaboration with content creators

Many educational content creators are well-recognized and actively engage with broader audiences. Science and maker content creators with any number of followers on social media platforms routinely produce content inspired by past or ongoing engineering research. Many who interact with the work of content creators will never see a peer-reviewed journal article on the topic but rely on content creators and influencers for information and creative inspiration. For our outreach, we identified and collaborated with content creators who studied and were familiar with science and engineering, engaged with and were well received by the maker community, had a relatively large following that reflected the target audience demographics, and which have a reputation that aligned with that of the research group. For collaboration with content creators we have found that interacting with creators that have one or multiple of these attributes, respective to the research area pursued, creates a large impact on the outreach of content. We first collaborated with Derek Muller to discuss compliant mechanisms and his YouTube video was posted March 12, 2019. Later we collaborated with Mark Rober in creating the world's smallest nerf gun; his video was posted September 30, 2023. With both content creators in this work, a mutually beneficial relationship was formed that resulted in the dissemination of accurate and ongoing research findings promoting broader impacts.

## Results

### Outputs

**Maker resources.** At the time of writing, we have uploaded fifty-six 3D printable designs to maker repositories (Thingiverse and Printables) and nine projects to Instructables. Six examples of these 3D printable models are shown in Fig 3. To support the model, we also uploaded lesson plans, descriptions, and short YouTube videos which complement the uploaded content.

**Video content creation.** In 2019, we identified and contacted several content creators with potential for collaboration. Derek Muller from the YouTube channel Veritasium (14.3M subscribers and 375 videos at the time of writing) agreed to the collaboration. The video "Why Machines That Bend Are Better" was released on March 12, 2019 [52]. The video featured an introduction to compliant mechanisms terminology, simplified explanations of the engineering principles involved, and demonstrations of compliant mechanisms. The CMR lab's Thingiverse profile was referenced in the video, and a link to resources was included in the video description. This made it possible for viewers to download and print their own models as seen

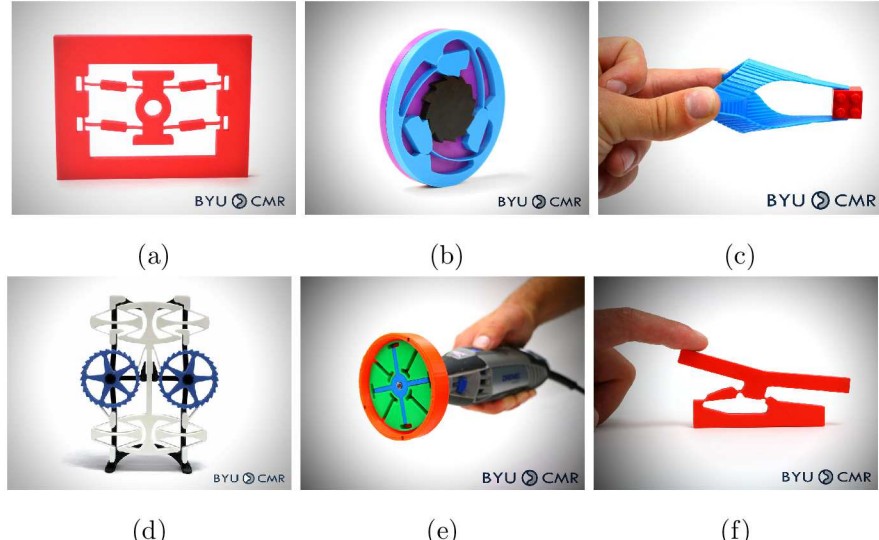

**Fig 3. Six examples of 3D printable models uploaded to online maker-resource repositories.** (a) Bistable Compliant Mechanism. (b) Compliant Overrunning Clutch. (c) Oriceps: Origami-Inspired Forceps. (d) Linear-Motion Compliant Mechanism. (e) Compliant Centrifugal Clutch. (f) Bistable Compliant Switch.

in the video. The primary data in this study are associated with this video because enough time has transpired to evaluate short, intermediate, and long-term impacts.

Four years later, the CMR lab was able to collaborate with YouTuber Mark Rober (26.6M subscribers and 126 videos at the time of writing). The work created over the course of this collaboration is posted in the video "World's Smallest Nerf Gun Shoots an Ant" on his YouTube channel [58]. The increased publicity again drove considerable traffic to the maker resources (which had at this point expanded to include both Thingiverse and Printables), lab website, and published research. Only the short-term outcomes of this video are available at the time of writing this paper, and are included to demonstrate repeatability.

## Outcomes

Short, intermediate, and long-term outcomes were measured to provide valuable insights into the overall impact of this outreach. First, the short, intermediate, and long-term outcomes are reported for the Veritasium collaboration. Then, short-term outcomes are also reported for the Mark Rober collaboration.

## Short-term outcomes

The short-term outcomes of this outreach are the immediate results that followed the project activities and outputs. They include measured data such as total reach, views, and downloads. Short-term outcomes represent the number of people that saw or took immediate action from the outputs.

**Content creation data.** Derek Muller published the video 'Why Machines That Bend Are Better' [52] on his YouTube channel, Veritasium, which accumulated over 11 million views in the first two years following the release. The increased publicity drove considerable traffic to the maker resources, lab website, and published research. Frequent spikes in traffic continue as more content creators, influencers, and individuals refer their followers to the outreach

content across various platforms (see Fig 6c). The average number of new users, views, and downloads from organic traffic remain noticeably higher than before the outreach began.

**Traffic and audience demographics.** Spread between the forty-one Thingiverse models posted before or in conjunction with the Veritasium video, nearly 600,000 views and 136,000 downloads were measured before September 2023. Additionally, over 35,000 views were counted between the eight models uploaded to Instructables. The number of downloads could not be measured internally on this platform. See 2 in 3 for a detailed view of the data pertaining to each uploaded model on Thingiverse and Instructables respectively.

Survey data revealed that 46% of visitors to the Thingiverse content described themselves as "hobbyists," 37% as "students," 12% as "professional engineers or designers", and 5% as "educators" (see Fig 4a). The hobbyist community is diverse, encompasses a wide range of interests and skill sets, and includes amateur engineers, small business owners, and DIY individuals.

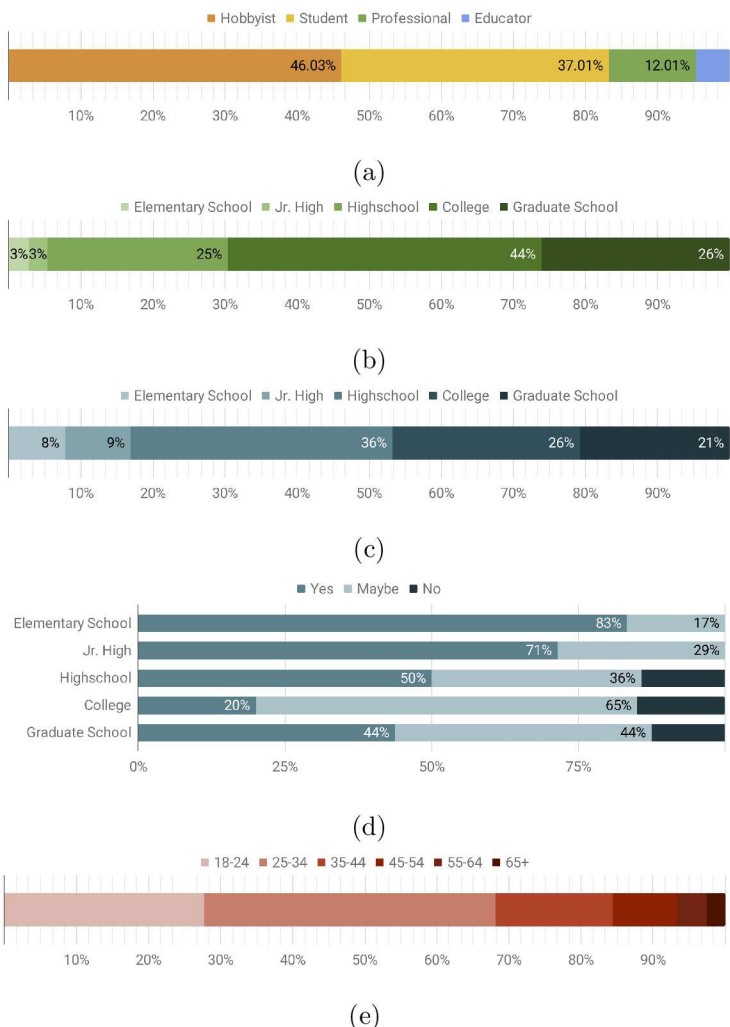

**Fig 4.** (a) Demographics of maker repository users that viewed and downloaded models. (b) Distribution of education level of those who declared themselves as students. (c) Distribution of education level of those who declared themselves as educators. (d) Spread of educators who plan on using the maker resources in the classroom. (e) Age distribution of website users.

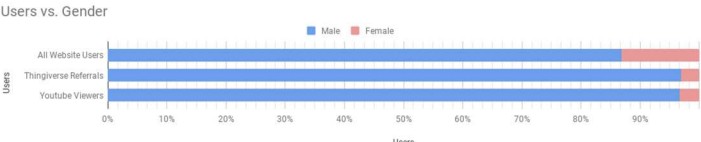

**Fig 5. Gender distribution based on new website users, Thingiverse referrals, and YouTube views.**

The education level of students ranged from 70% in undergraduate or graduate school, 25% in high school, and 6% in middle or elementary school (see Fig 4b). However, this and other dependent statistics may be biased by younger students being less likely to participate in a survey.

Educators represented 5% of all visitors. Educators are described as anyone involved with academics or facilitating the learning of others, including teachers, professors, librarians, school district directors, lab managers, and makerspace administrators. Fig 4c distinguishes the academic teaching level of participating educators and shows that 46% are undergraduate or graduate school professors, 36% are high school teachers, and 17% are middle or elementary school teachers. When asked if they intend to use the maker resources in the classroom, a reverse trend manifested with 83% of elementary school teachers, 44% of graduate school professors, and 20% of university undergraduate professors responding "yes" (see Fig 4d).

An age distribution (see Fig 4e) was obtained by using Google Analytics to track user traffic data from the research lab's website, which serves as a landing page for the outreach resources and additional research information. Most website users were between the ages of 18 and 34, with the number of users decreasing with age. A gender distribution of respondents is also shown in Fig 5. While there is a clear disparity, the gender breakdown is not unlike the current proportion typical of engineering. Current methods in use for decreasing the gender gap in STEM would also likely contribute to decreasing the gender gap in participants here.

CMR website traffic escalated as a result of the outreach activities and outputs. The source of new users shifted from primarily direct and organic searches to referrals (see Fig 6a). Site referrals originated from YouTube (88%), Thingiverse pages, and other social media platforms including Facebook, Pinterest, TikTok, Reddit, and online blogs and forums. Visitors navigated the website for additional outreach resources, supplemental research information, and citations to published content. This pattern carries over to Thingiverse users as shown in Fig 6b with 75% of new visitors referred to the maker resources through social media, 15% from browsing Thingiverse, 3.9% from organic google searches, 3.6% from the CMR lab website, 1.5% from word of mouth, and 1.1% from published research articles.

## Intermediate outcomes

The intermediate outcomes represent the secondary impacts that are triggered by short-term outcomes such as views and exposure.

**Intentions to act.** Respondents answered questions about their intended purpose of downloading the compliant mechanisms maker resources. Fig 7a groups the responses. The vast majority (79%) downloaded the models to satisfy their curiosity. This curiosity, combined with the hands-on, engaging element, makes this dissemination method unique from traditional methods such as published journal articles. Other intended uses include using the models for experimentation (63%), personal use (24%), as a teaching tool (8%), and implementing compliant mechanisms for professional use (7%).

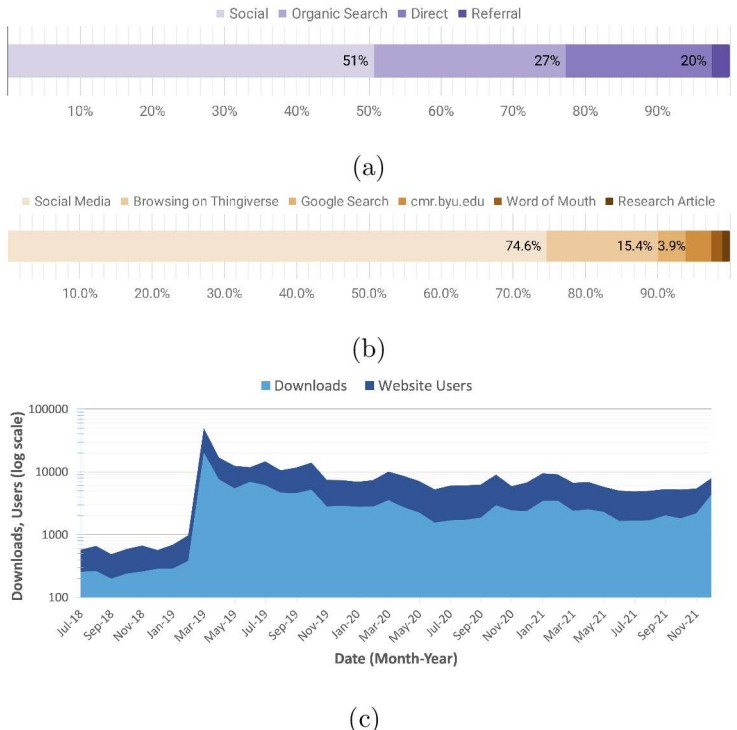

**Fig 6.** (a) Website user referral sources. (b) Maker repository user referral sources. (c) Maker repository download statistics overlaid with CMR website traffic from July 2018 to Dec 2021. Data is shown on a logarithmic scale.

**Social media branching.**   Other secondary online content creators and influencers rapidly began to reference the content found on Thingiverse and the Veritasium video. While Veritasium's video was the result of direct interaction with the CMR lab, all secondary videos were created and uploaded without any direct interaction. Numerous secondary videos uploaded to YouTube by other channels featured components of the outreach work (models provided on maker repositories or other resources), resulting in at least another 4 million accumulated YouTube views within 2 years. Additional video topics ranged from design integration of uploaded 3D models, personal experiences printing the models, review videos, and recorded class presentations.

Online blogs, forums, news sources, social media accounts, and other digital content creators also referenced resources on Thingiverse, sparking discussions on online forums and in the comments sections. Examples include the *Smithsonian Magazine* referencing the Oriceps model on Thingiverse in an online article titled "How Origami Is Revolutionizing Industrial Design" [53]. A significant spike of downloads occurred on the Oriceps model following the post. As more secondary and tertiary outcomes materialize, additional spikes continue to occur, as shown in Fig 6c.

More important than the rapid increase of traffic following newly published content from creators is how the updated rolling average of new users, views, and downloads is higher each time after the initial excitement has settled. Rather than a one-time boost, awareness and search engine optimization (SEO) increase permanently, branching to new possibilities of distribution across the internet, social media platforms, professional organizations, and unique audiences. Continually adding to the supply of new content on these channels initiates growth by providing resources for educational content creators.

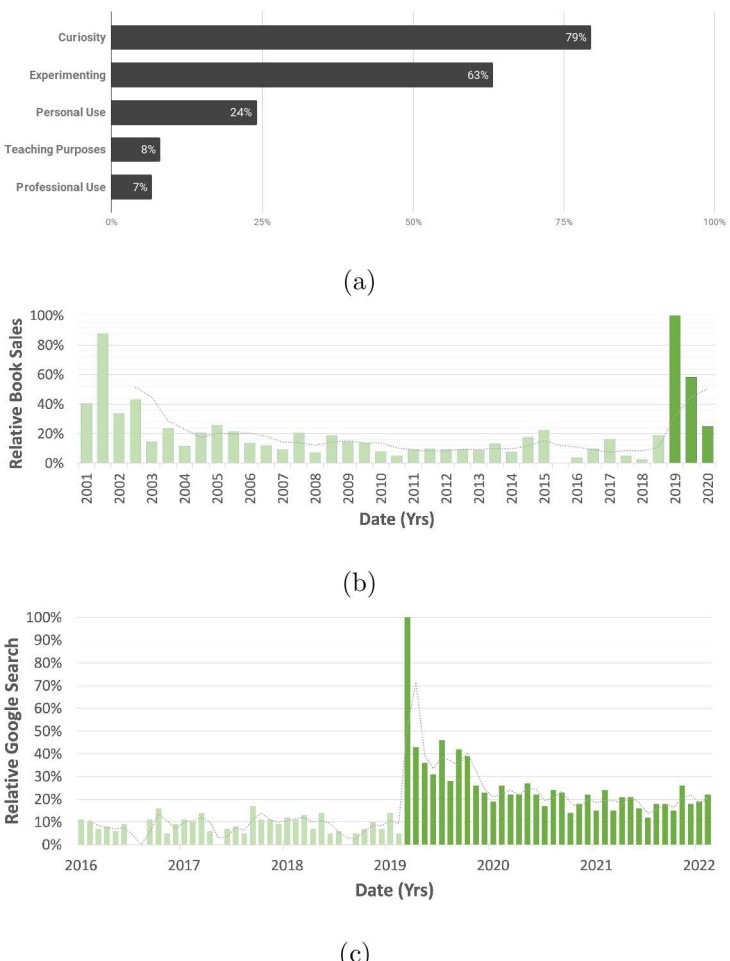

(a)

(b)

(c)

**Fig 7.** (a) The intended use of the compliant mechanisms outreach resources of those who downloaded models from Thingiverse. (b) Relative book distribution for *Compliant Mechanisms*. Book sales reported by the publisher, John Wiley & Sons, Inc. between 2001 and 2020. (c) Google search trends for "compliant mechanisms" between 2016 and 2022.

It is not feasible to track all branches and measure how they have made impacts, but identifiable ones indicate that the scale of impact is much larger than what can be directly linked. It only takes one maker (engineer, designer, educator, or student) to see a post to receive the inspiration to make a meaningful impact in our society. Posting content to online design-sharing platforms and working with content creators opens the door to increasing possibilities and impact.

**Inspire new applications.** Under a creative commons license, supplied 3D models are remixed and re-uploaded by members of the maker community which further innovation and research [54]. Beyond downloading models and adapting them for unique applications, makers and researchers also find inspiration to create their own compliant mechanism designs. For example, one YouTuber uploaded a video of his design for a complex compliant mechanism flexure joystick and throttle and shared the open-source design files on Thingiverse [55]. The video outlined his design, provided others with tutorials to recreate the innovation, and garnered over 40,000 views. The creator shared with us privately that his inspiration came from the maker resources that the CMR lab provided.

Outreach resources also prompted academic researchers to further the research. One example of this is a group of researchers who reverse-engineered and analyzed the compliant pliers uploaded to Thingiverse to determine reliability and functionality when using common polymers for 3D printer filaments such as ABS, PLA, and TPU [56].

**Increased textbook distribution.**   Increased *Compliant Mechanisms* book distribution is correlated to the outreach efforts, which strengthens the claim that the maker community is an audience that is receptive and eager to utilize more advanced supplementary information. As seen in Fig 7b, book distribution jumped more than 700% relative to the previous fifteen years, rivaling sales at the book's release, and corresponds to the release of material from content creators and engaging with the maker community.

**Consistent terminology usage.**   This outreach effort helped improve consistent accepted terminology among various communities. While select members of the academic community knew the technical term "compliant mechanism", some of the engineering community, makers and the general public may describe a compliant mechanism using ad hoc terminology, making it difficult to cross-reference other applicable material, including peer-reviewed publications. One way to track the adoption of this terminology is through the change in usage over time using search data. We analyzed Google search trends and found that searches using key terms, such as "compliant mechanisms", had increased from before the maker outreach was executed (see Fig 7c).

Social media platforms like Instagram that use hashtags have become a platform for makers to share their work and ideas on compliant mechanisms in centralized areas. The hashtag *#compliantmechanisms* was nearly obsolete before this outreach. Now there are hundreds of posts from users sharing their own pictures and videos of compliant mechanisms, many of which are direct prints from models the CMR lab uploaded to maker repositories.

**Opportunities for research funding.**   The breadth of influence of the outreach meant that people in many areas saw potential benefits for the research in their fields. While most will use the work to influence their products directly, others reach out for help; a small subset of the outreach audience is potential research sponsors. New projects and applications for our research thus arise from outreach, which accelerates the work to have a greater impact.

**Contributions by the maker community.**   Depending on the nature of the technology, a research group may have technical expertise but need more practical experience possessed by members of the maker community and industry. Makers and industry experts may have experience and perspectives that surpass that in the research lab when finding practical applications, prototyping, and manufacturing methods. The potential integration of online communities with new product development and platforms like Thingiverse generates an innovation community that provides "rapid-response and instantaneous feedback concerning different innovation projects throughout the entire innovation process" [57].

On several of our models posted on maker repositories, makers have commented, produced videos, posted ideas about how to improve the designs, shared their attempts at economical prototyping, and contributed other valuable insights.

## Long-term outcomes

Long-term outcomes consist of lasting impacts on broader communities due to the original outreach outputs and activities. The initial survey reported earlier indicated intentions to act, while a second follow-up survey helped determine if and how the individuals followed through two to three years following the initial exposure. This follow-up data indicates if any systematic behavior change has occurred. Of the 2,050 visitors who filled out the initial survey, 1,183 (58%) provided their email addresses to learn about compliant mechanisms or provide

additional feedback. Of the provided contact information, 96% were valid emails; 48% of email recipients opened the follow-up message, and roughly 12% proceeded to fill out the brief survey.

**Sustained familiarity with technical topic.**    Similar to the first survey, responses to this second survey included students, educators, professional engineers and designers, and hobbyists. The survey questioned Thingiverse visitors how their familiarity with compliant mechanisms has changed since they were first exposed to the outreach content more than two years earlier. As shown in Fig 8a, over half of the respondents (51%) answered that they were "not at all familiar" with compliant mechanisms prior to discovering the outreach content. Meanwhile, two years later, a significant shift towards familiarity has been maintained, with the majority (95%), declaring that they are currently "somewhat" or "very familiar" with compliant mechanisms. The remaining (5%) stated they are only "slightly familiar". This data suggests that the outreach efforts did more than bring temporary exposure but made a lasting impression with retained knowledge years later to broader audiences.

**Influenced to act or think in new ways.**    The outreach triggered audiences to act or think in new ways. Respondents proactively sought additional sources to increase their familiarity

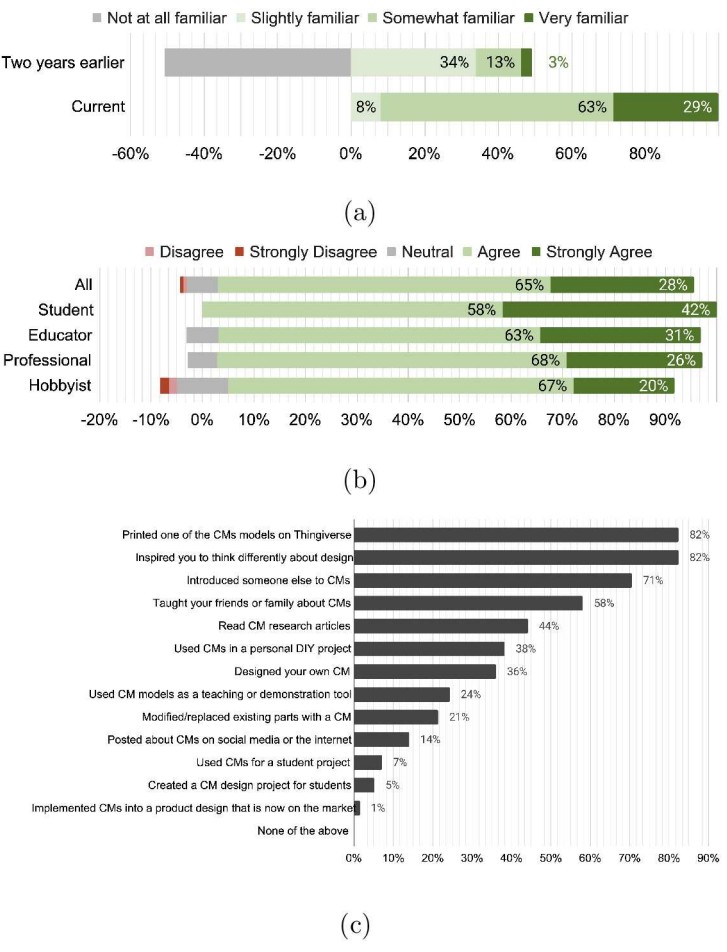

(a)

(b)

(c)

**Fig 8.**    (a) Familiarity with the technical topic, compliant mechanisms, before discovering the outreach content on Thingiverse and now. (b) Thingiverse users agreement with the following statement: "The compliant mechanisms models and resources influenced you to act or think in new ways." (c) Responses from Thingiverse users indicating what they have done differently as a result of finding the compliant mechanisms outreach resources.

and confidence in employing compliant mechanism design principles for practical applications. Sixty-three percent of respondents agreed, and 29% strongly agreed that the models and resources influenced them to act or think in new ways (Fig 8b). Fig 8c categorizes and ranks "how" and "what" they have done. Eighty-four percent of respondents downloaded at least one of the compliant mechanism models on Thingiverse with 38% noting that they also created original designs, including ideas for personal DIY projects and commercialized products.

**Referring others.** Three out of four survey respondents indicated that they introduced someone to compliant mechanisms. Methods of sharing, teaching, or making an impression on someone using compliant mechanisms included word-of-mouth discussions with colleagues, friends, and family (84%), posting on social media or an online blog (19%), student presentations (14%), work presentations (12%), and class lectures (9%) (see Fig 9a). Evidence of primary visitors referring others to the outreach resources or compliant mechanism information became apparent through repeated spikes in visitor and model download traffic and prolonged increase of search trends.

Respondents shared estimates of how many people have seen or heard something they have said, posted, or shared about compliant mechanisms. Fig 9b plots the data on a logarithmic scale showing the relationship between the percentage of Thingiverse visitors with the number of referrals they provided. The average number of referrals reported was 48.5 per person with a median and mode of 10.

Fig 10a displays the teaching level of educators who used compliant mechanisms outreach material for class lectures, student design projects, or as a teaching or demonstration tool. High school-level educators used the material the most often, with makerspace or lab advisors coming in second. Fig 10b breaks down the distribution of the various education levels of students that used compliant mechanisms for student presentations. Seventy-five percent of student presentations from survey respondents were from university undergraduate or graduate students, and the remaining twenty-five percent were from high school students.

**Other materials referenced.** Fig 11 categorizes the most common resources respondents reported referencing to seek supplemental knowledge about compliant mechanisms within the

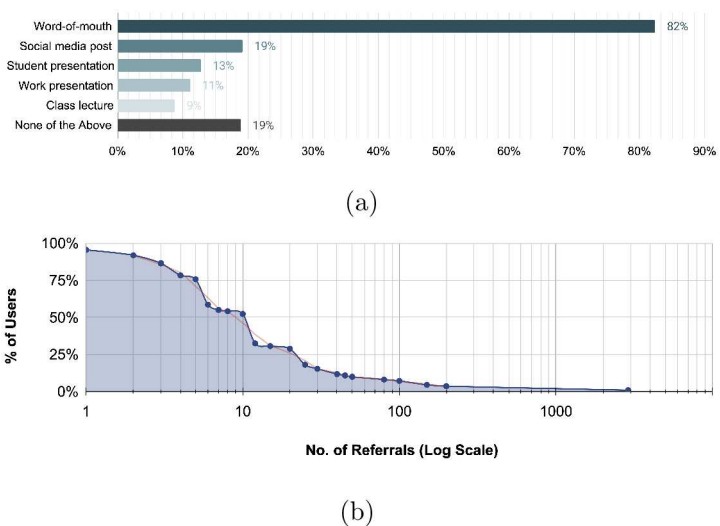

(a)

(b)

**Fig 9.** (a) Referral methods carried out by Thingiverse users. (b) Estimate of how many people have seen or heard something respondents have said, posted, or shared about compliant mechanisms. (e.g., talked to, taught, saw a social media post, watched a YouTube video, listened to a presentation, read a blog post, etc.). The data is plotted on a logarithmic scale on the x-axis for the number of referrals.

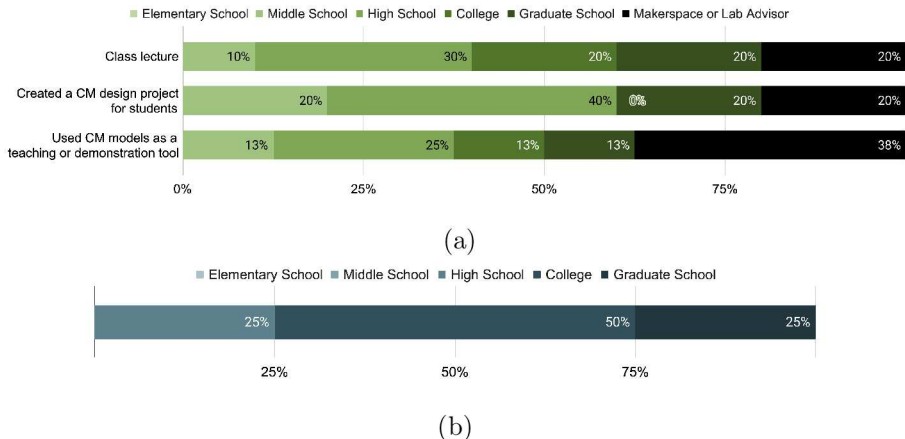

(a)

(b)

**Fig 10.** (a) Distribution of the various teaching levels of educators who used the compliant mechanism outreach material for class lectures, student design projects, or a teaching/demonstration tool. (b) Distribution of the various education levels of students who used the compliant mechanism outreach material for student presentations.

two years after seeing the outreach materials. Seventy percent of respondents searched for compliant mechanisms on social media sites, 50% discovered online articles and blogs, 15% obtained published journal or conference articles, 11% searched for online lectures, 10% bought or referenced the *Handbook of Compliant Mechanisms*, 6% bought or referenced the textbook *Compliant Mechanisms*, and 2% took a compliant mechanisms course. This data highlights that engaging broader audiences is an effective way to drive traffic and utilize published content. It also indicates that social media is a prevailing method that broader audiences rely on for discovering additional resources about an unfamiliar technical subject.

## Repeated study

The process of creating maker materials and collaborating with an internet content creator was repeated with engineering and STEM internet influencer Mark Rober. The Mark Rober YouTube channel had over 26 million subscribers and over 125 videos at the time of writing. The video "World's Smallest Nerf Gun Shoots an Ant" was released September 30, 2023 [58]. The video featured a novel compliant mechanism design and provided an overview of the

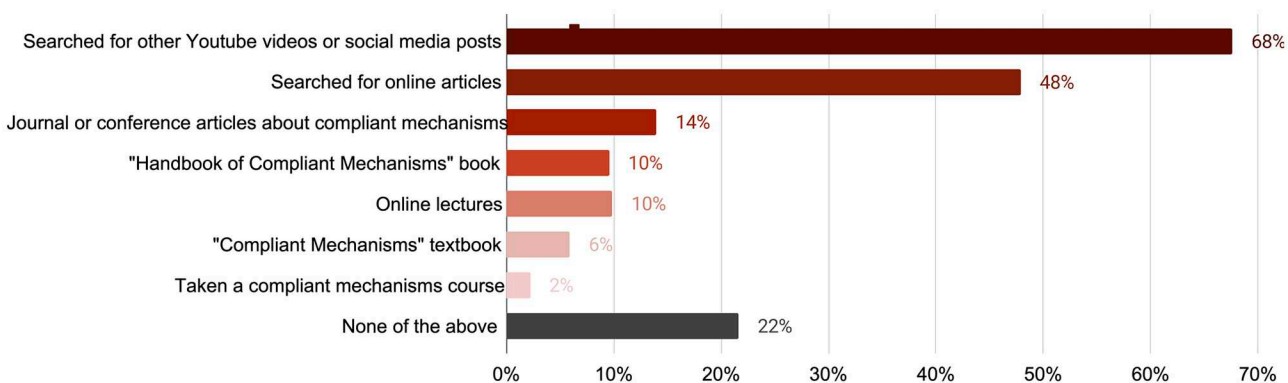

**Fig 11. Other compliant mechanisms resources and learning material used or referenced by Thingiverse survey respondents.**

general technology. A reference to the CMR lab's available maker resources was made in the video, and a link to the resources was included in the video description. The 3D printable models made previously were still available, but four new models were introduced specifically related to this video, as shown in Fig 12. One of these, the compliant blaster (Fig 12a, is the device that is a focus of the video (a one-piece compliant mechanism that can be built at multiple size scales). Second (Fig 12b) was a micromanipulator used in the project but not mentioned, and third (Fig 12c) was a Lego-compatible compliant mechanism not used on the project but mentioned in the video. Last, the compliant disk launcher (Fig 12d) uses the same basic components as the blaster, but could be more useful for educators because it looks less like a weapon. This device was not mentioned in the video but was available on the repositories.

Each of these four models were posted immediately after the video. Table 1 shows the number of views and downloads for each model, compared to whether it was mentioned in the video and used in the project. The differences in these numbers allows us to isolate the impact of a model being mentioned in the video, used in the project, or simply being posted on the repository at the time of increased traffic. It is likely that the model which was neither used in the project nor mentioned in the video (the compliant disk launcher) received comparatively such a high number of views and downloads because of its similarity to the compliant blaster.

The Mark Rober video received 24 million views in the first month after its release. It was translated into 10 languages and views spanned dozens of countries. The four 3D printable models received a total of 363,378 views and 96,973 downloads on public repositories

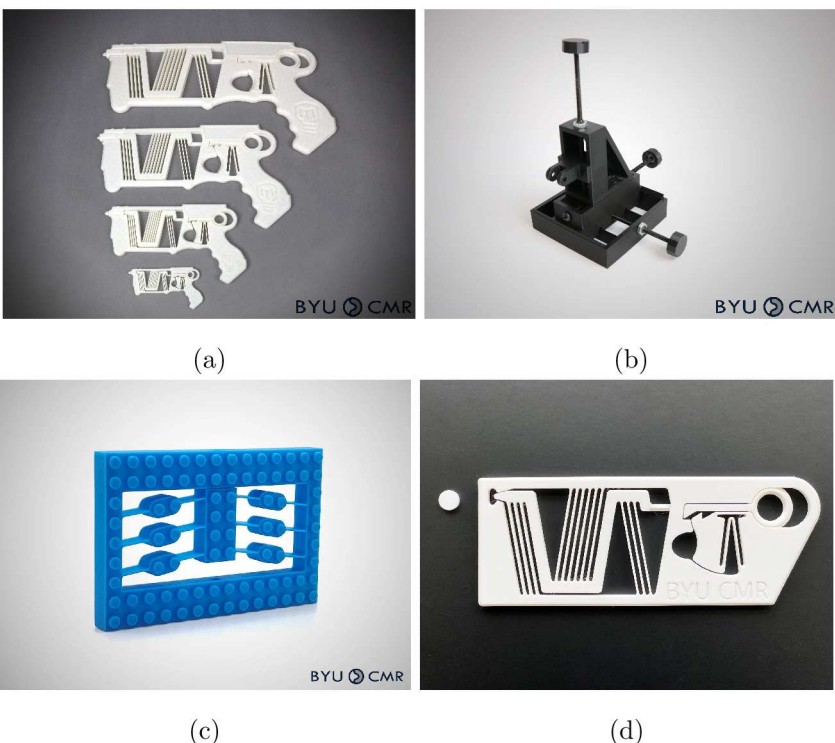

(a)

(b)

(c)

(d)

**Fig 12. The four new 3D printable models introduced to maker repositories.** (a) One-Piece Compliant Mechanism Blaster. (b) Micromanipulator. (c) Lego-Compatible Bistable Mechanism. (d) One-Piece Compliant Mechanism Disk Launcher.

**Table 1. Relationship of newly released models and the second video.**

| Used in Project | | Mentioned in Video | |
|---|---|---|---|
| | | **Yes** | **No** |
| Yes | Fig | 12a | 12a |
| | Views | 300,750 | 10,021 |
| | Downloads | 86,669 | 344 |
| No | Fig | 12c | 12d |
| | Views | 6,827 | 45,780 |
| | Downloads | 732 | 7,228 |

(Thingiverse and Printables) measured 25 days after video release. Table A in S1 File shows the number of downloads for each.

The 52 models available previous to this video had received a total of 511,738 views and 147,108 downloads since the maker space profiles were created and received an additional 37,423 views and 1,773 downloads during the 25 days following the release of the second video.

The CMR lab website saw an increase in viewership, with over 5,000 visits in the day the video was posted and an additional 15,000 in the following three weeks. Social media branching also occurred with other online content creators and influencers referencing the video and maker content. Among other topics, secondary videos included modifications and critiques of the maker resources provided. The terminology of compliant mechanisms also continued to spread to broader audiences. Google search trends were again analyzed and are seen in Fig 13a. "Compliant mechanisms" and related search terms increased after the video by 375%.

Book distribution is another measure of extending technical knowledge to larger audiences. Although official sales data on the related book were not yet available at the time of this writing (Oct 2023), there are indications that more people are learning from that resource also, such as being ranked in the top ten selling books on Amazon.com for the Mechanical Engineering topic area and the more general class of Engineering.

These short-term outcomes (maker resource views and downloads, Google searches, and lab website visits) mirror those experienced in the first instance, and suggest that similar intermediate and long-term outcomes will also emerge with time. It is also important to note that COVID-19, which began to initiate international lockdowns in early 2020, significantly increased the activity of people looking for content, including makers looking for models to create. This trend towards making has continued through the ensuing four years. This may be a factor influencing the greater numbers seen following the second video.

## Discussion

This case study demonstrates extending the impact of engineering research to broader communities by engaging the maker community and collaborating with digital content creators. Principal findings from this study include:

1. Broader communities are receptive to research findings when communicated through contemporary dissemination channels. They are able to apply their new knowledge to make a difference.

2. The maker community is a professionalizing field of collaborative individuals, including engineers and designers, students, educators, researchers, and skilled hobbyists.

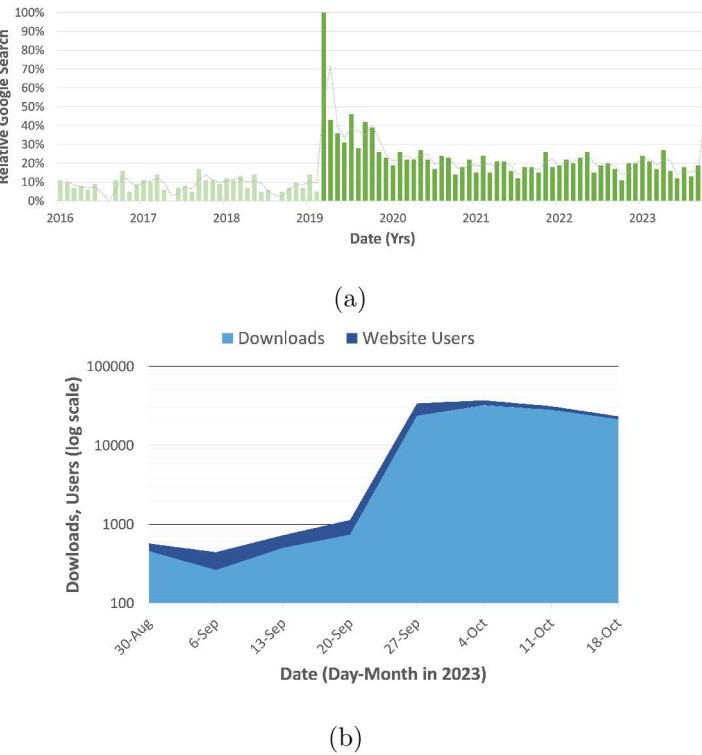

**Fig 13.** (a) Google searches from 2016 to Oct 2023. (b) Maker repository download statistics overlaid with CMR website traffic from Aug 2023 to Oct 2023. Data is shown on a logarithmic scale.

3. Collaboration with content creators catalyzes the dissemination of research results to interest individuals.

4. Online design-sharing platforms are effective channels to reach the maker community.

5. Leveraging contemporary distribution channels (e.g., social media, collaborating with content creators, online model repositories, and blogs) complements traditional academic distribution channels (e.g., academic papers, textbooks).

6. Improving general public awareness of terminology promotes increased organic traffic to existing and new content on a subject.

7. Academic research groups, citizen scientists, and industry professionals can benefit from insights from the maker community, such as prototyping methods and design applications.

8. Engaging broader audiences with hands-on learning experiences, such as downloading and 3D printing an engineering model, is a strong driver to retain new knowledge and inspire individuals to think and act in new ways.

9. The maker community is proactively referring others to technical resources.

Results from the first case study include measured short-term, intermediate, and long-term outcomes. Short-term outcomes, measurable across both case studies, showed that social media was the most effective referral method to maker resources and lab website landing pages. Intermediate outcomes for the first case study revealed the organic behavior of other individuals and content creators disseminating the outreach content to their network of

followers resulting in a sustained increase in traffic. Secondary outcomes also suggest the complementary nature of mixing contemporary and traditional distribution channels. Increased engagement with the broader communities increased textbook dissemination and readership of peer-reviewed published content. A sustained shift in public awareness and search behavior using established terminology was also shown by analyzing Google Search trends over the past seven years. Long-term outcomes for the first case study were measured two and a half years after the outreach initiatives began, and displayed a sustained knowledge base among outreach recipients. Though not discussed in detail here, secondary long-term outcomes also suggest that increased exposure among broader audiences can lead to additional research sponsors.

Numerous examples show how the maker community uses the newly-found research knowledge from the outreach initiative for innovative applications and educational purposes. These makers then share their ideas on social platforms resulting in a perpetual growth cycle of hundreds, thousands, or millions of impressions among diverse communities. This readily available content curated for a wide range of demographics helps the next generation of students, educators, engineers, designers, and hobbyists to be more informed, share ideas, and implement novel applications in creative and timely ways. As academic research-inspired work grows on social platforms, the credibility and public awareness of the research group also can increase.

These outreach case studies have important implications for the academic and engineering research communities. The data suggests that the preferences of relevant audiences to quickly find new knowledge prioritizes contemporary dissemination channels (e.g., social media) over traditional methods (e.g., peer-reviewed publications). Research labs can adopt these distribution channels into their standard practices to ensure the continued distribution of high-quality, accurate, and relevant research findings. The concepts presented here are easily applicable to other areas of STEM. Biologists might make available models of cross-sections of a cell of interest, or paleontologists fossils relevant to their research, or mathematicians geometric representations of the theorems they research. Additionally, while this framework was developed using a US-based research lab, it is expected that similar strategies would be successful in other countries.

In a university research environment, this type of outreach work has additional benefit: it presents a way for newer students to be involved in making an immediate impact, while helping them gain a more in-depth understanding of the research area. This prepares them for more advanced work.

Limitations to this outreach initiative include the limited research labs that precisely match the niche characteristics to follow this framework. The Compliant Mechanisms Research Group at Brigham Young University produces mechanical hardware conducive to this framework. However, the general principles of engaging the maker community and content creators with hands-on learning opportunities are widely applicable with comparable impact potential. Additional limitations of the case studies include potentially biased results from a survey sample size that only partially represents the entire population of maker repository visitors. It is assumed that the observed sample size inherently includes a higher percentage of individuals more passionate about compliant mechanisms.

## Conclusion

An evolving media landscape and growing public reliance on social and collaborative platforms to spread and discover knowledge offer researchers an opportunity to take advantage of contemporary dissemination strategies to broaden the impact of their research. The methods presented in this case study demonstrate the potential to accelerate engineering research

distribution with scientific and broader communities. Collaborating with STEM-oriented digital content creators and using online design-sharing platforms to upload 3D models and educational resources such as instructional videos, lesson plans, and tutorials can supplement traditional publication methods by allowing broader audiences to access, learn, and replicate the work. While limitations exist, these outreach practices can be valuable tools for engineering researchers to extend the impact of their research by providing novel and engaging resources for education and outreach.

## Supporting information

**S1 File. Fig A and Tables A-B. Figure A**. Outcomes-focused project logic model which includes *1) Rationale/Needs*: Our desired vision and what we hope our actions will result in, *2) Inputs*: People, time, funds, and other resources dedicated to the success of the project, *3) Activities*: The actions that are taken to achieve desired results, *4) Outputs*: tangible direct products of project activities, and *5) Outcomes*: Impacts and changes expected as a result of the project. **Table A**. Total downloads and views statistics for the fifty-seven 3D-printable models provided by maker repository analytics, retrieved the week of October 22, 2023. **Table B**. Total views statistics for the eight "Instructables" provided by Instructables analytics.
(ZIP)

**S1 Dataset. Raw data from survey results.**
(ZIP)

## Acknowledgments

We thank Dr. Dennis Eggett from the Brigham Young University Department of Statistics Consulting Center for providing direction to accurately represent the data from this case study. We also thank Derek Muller and Mark Rober and his team for their collaboration and support as excellent STEM influencers and digital content creators. Various undergraduates from the CMR lab also contributed to the outreach materials, included Davis Wing, Trevor Carter, Audrey Christiansen, Lydia Beazer, and Amanda Lytle Bartschi.

## Author Contributions

**Conceptualization:** Jacob L. Sheffield, Terri Bateman, Spencer Magleby, Larry L. Howell.

**Formal analysis:** Jacob L. Sheffield, Aliya Bascom.

**Funding acquisition:** Jacob L. Sheffield, Larry L. Howell.

**Investigation:** Jacob L. Sheffield.

**Methodology:** Jacob L. Sheffield.

**Supervision:** Terri Bateman, Spencer Magleby, Larry L. Howell.

**Visualization:** Jacob L. Sheffield, Aliya Bascom.

**Writing – original draft:** Jacob L. Sheffield.

**Writing – review & editing:** Bethany Parkinson, Terri Bateman, Spencer Magleby, Larry L. Howell.

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
