## [Decision Letter · Decision Letter 0]

13 Feb 2024

PONE-D-23-38190Expanding research impact through engaging the maker community and collaborating with digital content creatorsPLOS ONE

Dear Dr. Parkinson,

Thank you for submitting your manuscript to PLOS ONE. After careful consideration, we feel that it has merit but does not fully meet PLOS ONE’s publication criteria as it currently stands. Therefore, we invite you to submit a revised version of the manuscript that addresses the points raised during the review process.

We look forward to receiving your revised manuscript.

Kind regards,

Van Thanh Tien Nguyen

Academic Editor

PLOS ONE

Journal Requirements:

"This work was made possible by the National Science Foundation (NSF) through Award No. 1663345, “Mechanisms on Developable Surfaces”, which included as part of its Broader Impacts component a task to engage the maker community. We also thank Dr. Dennis Eggett from the Brigham Young University Department of Statistics Consulting 

Center for providing direction to accurately represent the data from this case study. We also thank Derek Muller and Mark Rober and his team for their collaboration and support as excellent STEM influencers and digital content creators. Various undergraduates from the CMR lab also contributed to the outreach materials, included Davis Wing, Trevor Carter, Audrey Christiansen, Lydia Beazer, and Amanda Lytle Bartschi."

"This work was made possible by the National Science Foundation (NSF, https://www.nsf.gov/) through Award No. 1663345, "Mechanisms on Developable Surfaces", awarded to LLH. This award included as part of its Broader Impacts component a task to engage the maker community. The sponsors did not play any role in the study design, data collection and analysis, decision to publish, or preparation of the manuscript."

**Additional Editor Comments:**

Dear Dr. Bethany Parkinson:

Thank you for submitting your interesting manuscript to PLOS ONE.

We are pleased to announce that We have received 3 positive comments and suggestions for your manuscript; therefore, We have now made a decision of "Minor Revision" for your manuscript.

Please revise your manuscript following the reviewers' comments in the review reports section below.

By this time, you have a chance to improve your manuscript in both its contents and English style.

We are looking forward to receiving your resubmission.

Please do not hesitate to contact us if you have further questions or concerns.

Again, thank you for your time and contributions.

Best regards,

Van Thanh Tien Nguyen, Ph.D.

Academic Editor,

PLOS ONE, SCIE Q2/Scopus Q1 (2022)

Reviewers' comments:

Reviewer's Responses to Questions

**Comments to the Author**

1. Is the manuscript technically sound, and do the data support the conclusions?

Reviewer #1: Yes

Reviewer #2: Partly

Reviewer #3: Yes

2. Has the statistical analysis been performed appropriately and rigorously? 

Reviewer #1: Yes

Reviewer #2: Yes

Reviewer #3: Yes

3. Have the authors made all data underlying the findings in their manuscript fully available?

Reviewer #1: Yes

Reviewer #2: No

Reviewer #3: Yes

4. Is the manuscript presented in an intelligible fashion and written in standard English?

Reviewer #1: Yes

Reviewer #2: Yes

Reviewer #3: Yes

5. Review Comments to the Author

Reviewer #1: This article proposes a method to increase the impact of academic research by making the material available for public use, thereby engaging the maker community and collaborating with internet content creators to expand reach. This is an approads intersting. The research process took place over four years with short-term and long-term data collection, showing that data collection was taken seriously. The research results are analyzed in detail. The article is suitable for publication.

Reviewer #2: The paper reports on a hugely successful initiative to engage the maker community with research by professional and academic engineers. The authors set out a framework for others to follow in order to emulate their success. I thought the paper was well written, well founded and the display items were neatly presented. I fully agree with the sentiment of the paper that there’s a mutual benefit to be had by pushing at what’s effectively an open door with an already interested public. I do have some comments and suggestions that I set out below for consideration by the authors.

Figure 1. Here you set out your linear framework for dissemination, but I wonder if this speaks to an older form of science communication (the deficit model) that doesn’t accurately reflect what you found. I could see the figure working better as a cycle because in engaging with the community you’re seeding new academic researchers who will conduct new research and start the cycle over. Thus, it’s far more participatory. See ‘Trench, B. (2008). Towards an analytical framework of science communication models. Communicating science in social contexts: New models, new practices, 119-135.’

With a view of how this could be extended to other research communities, I found parts of the paper quite vague:

1. Line 187 you write: “Considering the turnover in a typical university research lab, tutorials and templates were created to outline a repeatable and sustainable process to help the next generation of students continue supporting the ongoing outreach strategy” but what are these tutorials and templates about?

2. Line 199 you write how “we identified and collaborated with content creators who (1) were adequately informed in their area of research, (2) were engaging and well-received by the desired community, (3) had a substantial following that reflected the target audience demographics, and (4) had a reputation that aligned with that of the research group.” None of these are defined, what’s a substantial following for example. Is there a threshold for this? How do you assess someone’s reputation? All valuable information for your reader. Clearly, the content creators are incredibly popular and fit these criteria but, as it stands, this sounds like a post hoc rationalization rather than something others could use.

Methods – please add some times of the project interventions e.g. the dates the YouTube videos of the content creators went live.

In light of open science principles, are the data from the surveys and code for your figures available somewhere?

Line 140 - For the models you created you list some considerations that would make them more likely to be adopted by the maker community. Did you measure which of the considerations each of the objects met? Perhaps those that met specific criteria were more likely to be used - something you could test statistically.

At some points I found your use of the passive voice a bit confusing because it masked who was responsible for the action. E.g. “At the time of writing, fifty-six 3D printable designs have been uploaded to maker repositories (Thingiverse and Printables) and nine projects have been uploaded to Instructables. Six examples of 3D printable models uploaded to Thingiverse and Printables are shown in Fig 3.”

You focus your data analysis on descriptive statistics rather than anything inferential, so you’re ultimately left with correlations between your initiatives and the various measures of popularity. Though it’s clear your work was driving these patterns I think you could consider discussing how to quantify your impact. Some recent papers describe how to do so e.g. the causal impact framework Brodersen KH, Gallusser F, Koehler J, Remy N, Scott SL. Inferring causal impact using Bayesian structural time-series models. Annals of Applied Statistics, 2015, Vol. 9, No. 1, 247-274. http://research.google.com/pubs/pub41854.html Just to be clear, I’m not hinging my review on this point but it should be acknowledged especially given the confound that COVID may have had on some these findings with an eager maker community stuck at home and looking for content.

Figure 11. is the order the wrong way around here? Panel (c) looks like the Lego-Compatible device not panel (d). And when I check out this site (https://compliantmechanisms.byu.edu/maker-resources) that you refer to earlier panel (d) looks like a ‘One-Piece Compliant Disk Launcher’

In the discussion and even the introduction I think you could offer your perspective on how the framework could be expanded to other areas in STEM. Are there some fields where it’s easy to apply (biology/ palaeontology, which is my own area, seems like an obvious one – imagine having a 3D model of a new dinosaur fossil made available)? Or is there something unique about the maker community? Further, you don’t mention citizen science in your work, but it’s intimately related to these ideas.

More could be made of the limitations / future directions of the work. Is this type of framework applicable to other countries? Is there a gender component? Are we beholden to hugely popular content creators to catalyze dissemination? How do we encourage more people to take on these sorts of roles when academia is so invested in prioritizing research funding and publications over engagement? I’m conscious that you can’t consider all of these points, but they all came to mind as I was reading the piece.

Some of the references are missing links to the relevant source e.g. ‘Green T. Publication is not enough, to generate impact you need to campaign. Impact of Social Sciences. 2019.’

On a final note, I watched both videos and now know what compliant mechanisms are, so you’ve got through to another person!

Adam Kane, University College Dublin

Reviewer #3: The manuscript is technically sound, and the data support the conclusions.

The statistical analysis has been performed appropriately and rigorously.

The authors have made all data underlying the findings in their manuscript fully available.

The manuscript is presented in an intelligible fashion and written in standard English.

6. PLOS authors have the option to publish the peer review history of their article (what does this mean?). If published, this will include your full peer review and any attached files.

Reviewer #1: No

Reviewer #2: **Yes: **Adam Kane

Reviewer #3: No

---

## [Author Response · Author response to Decision Letter 0]

21 Mar 2024

Please see the attached "Response to Reviewers."

---

## [Decision Letter · Decision Letter 1]

4 Apr 2024

Expanding research impact through engaging the maker community and collaborating with digital content creators

PONE-D-23-38190R1

Dear Dr. Parkinson,

We’re pleased to inform you that your manuscript has been judged scientifically suitable for publication and will be formally accepted for publication once it meets all outstanding technical requirements.

Kind regards,

Van Thanh Tien Nguyen, Ph.D.

Academic Editor

PLOS ONE

Additional Editor Comments (optional):

Reviewers' comments:

Reviewer's Responses to Questions

**Comments to the Author**

1. If the authors have adequately addressed your comments raised in a previous round of review and you feel that this manuscript is now acceptable for publication, you may indicate that here to bypass the “Comments to the Author” section, enter your conflict of interest statement in the “Confidential to Editor” section, and submit your "Accept" recommendation.

Reviewer #4: All comments have been addressed

2. Is the manuscript technically sound, and do the data support the conclusions?

Reviewer #4: Yes

3. Has the statistical analysis been performed appropriately and rigorously? 

Reviewer #4: Yes

4. Have the authors made all data underlying the findings in their manuscript fully available?

Reviewer #4: Yes

5. Is the manuscript presented in an intelligible fashion and written in standard English?

Reviewer #4: Yes

6. Review Comments to the Author

Reviewer #4: Dear Editor and Authors:

Thank you for providing the point to point response.

All concerns have been solved and the revised manuscript has been improved significantly.

It is ready for publication.

Thank you for reading.

7. PLOS authors have the option to publish the peer review history of their article (what does this mean?). If published, this will include your full peer review and any attached files.

Reviewer #4: No
